# ENERGY-BASED VIEW OF RETROSYNTHESIS

## ABSTRACT

Retrosynthesis is the process of identifying a set of reactants to synthesize a target molecule. It is of vital importance to material design and drug discovery. Existing machine learning approaches based on language models and graph neural networks have achieved encouraging results. However, the inner connections of these models are rarely discussed, and rigorous evaluations of these models are largely in need. In this paper, we propose a framework that unifies sequence- and graph-based methods as energy-based models (EBMs) with different energy functions. This unified view establishes connections and reveals the differences between models, thereby enhancing our understanding of model design. We also provide a comprehensive assessment of performance to the community. Moreover, we present a novel dual variant within the framework that performs consistent training to induce the agreement between forward- and backward-prediction. This model improves the state-of-the-art of template-free methods with or without reaction types.

## 1 INTRODUCTION

Retrosynthesis is a critical problem in organic chemistry and drug discovery (Corey, 1988; 1991; Segler et al., 2018b; Szymkuć et al., 2016; Strieth-Kalthoff et al., 2020). As the reverse process of chemical synthesis (Coley et al., 2017a; 2019), retrosynthesis aims to find the set of reactants that can synthesize the provided target via chemical reactions (Fig 1). Since the search space of theoretically feasible reactant candidates is enormous, models should be designed carefully to have the expression power to learn complex chemical rules and maintain computational efficiency.

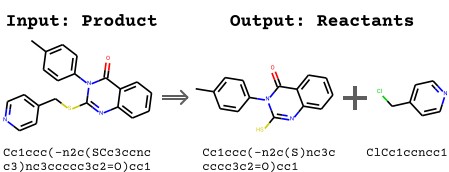

Figure 1: Retrosynthesis and SMILES.

Recent machine learning applications on retrosynthesis, including sequence- and graph-based models, have made significant progress (Segler & Waller, 2017a; Segler et al., 2018b; Johansson et al., 2020). Sequence-based models treat molecules as one-dimensional token sequences (SMILES (Weininger, 1988), bottom of Fig 1) and formulate retrosynthesis as a sequence-to-sequence problem, where recent advances in neural machine translation (Vaswani et al., 2017; Schwaller et al., 2019) can be applied. In this principle, the LSTM-based encoder-decoder frameworks and, more recently, transformer-based approaches have achieved promising results (Liu et al., 2017; Schwaller et al., 2019; Zheng et al., 2019). On the other hand, graph-based models have a natural representation of human-interpretable molecular graphs, where chemical rules are easily applied. Graph-based approaches that perform graph matching with chemical rules ("templates"; definition in Sec 3.2) or reaction centers have reached encouraging results (Dai et al., 2019; Shi et al., 2020). In this paper, we focus on one-step retrosynthesis, which is also the foundation of multi-step retrosynthesis (Segler et al., 2018b).

Our goal here is to provide a unified view of both sequence- and graph-based retrosynthesis models using an energy-based model (EBM) framework. It is beneficial because: First, the model design with EBM is very flexible. Within this framework, both types of models can be formulated as different EBM variants by instantiating the energy score functions into specific forms. Second, EBM provides principled ways for training models, including maximum likelihood estimator, pseudo-likelihood, etc. Third, a unified view is critical to provide insights into different EBM variants, as it is easy to extract commonalities and differences between EBM variants, understand strengths

and limitations in model design, compare the complexity of learning or inference, and inspire novel EBM variants. To summarize our contributions:

- We propose a unified energy-based model (EBM) framework that integrates sequence- and graph-based models for retrosynthesis. To our best knowledge, this is the first effort to unify and exploit inner connectivity between different models.

- We perform rigorous evaluations by running tens of experiments on different model designs. We believe revealing the performance to the community contributes to the development of retrosynthesis models.

- Inspired by such a unified framework, we propose a novel dual EBM variant that performs consistent training over forward and backward prediction directions. ~~dual model improves the state-of-the-art accuracy by 9.9% for full automate template-free and 2.7% for template-based~~.

The goal of this paper is to investigate the performance of different models under the setup without any hand-crafted chemistry features, e.g. reaction center, during training. Incorporating these hand-crafted chemistry features usually can boost accuracy significantly regardless of the model design. So adding features is not the focus of our paper. See discussion in Appendix A.2, A.3, A.4.

## 2 ENERGY-BASED MODEL FOR RETROSYNTHESIS

Retrosynthesis is to predict a set of reactant molecules from a product molecule. We denote the product as $y$, and the set of reactants predicted for one-step retrosynthesis as $X$. The key for retrosynthesis is to model the conditional probability $p(X|y)$ (Dai et al., 2019; Shi et al., 2020; Liu et al., 2017; Schwaller et al., 2019). EBM provides a common theoretical framework that can unify many retrosynthesis models, including but not limited to existing models.

An EBM (LeCun et al., 2006; Hinton, 2012) defines the distribution using an energy function. Without loss of generality, we define the joint distribution of product and reactants as follows:

$$p_\theta(X, y) = \frac{\exp(-E_\theta(X, y))}{Z(\theta)} \quad (1)$$

where the partition function $Z(\theta) = \sum_y \sum_X \exp(-E_\theta(X, y))$ is a normalization constant to ensure a valid probability distribution. Since the design of $E_\theta$ is free of choice, EBMs can be used to unify many retrosynthesis models by instantiating the energy function $E(\theta)$ with different designs and approximation of the partition function. Note there is a trade-off between model expression capacity and learning tractability. EBM is also easy to obtain arbitrary conditioning with different partition functions. For example,

**Algorithm 1** EBM framework

**[Train Phase]: Learning**
**Input**: Reactants $X$ and products $y$.
**1.** Parameterize $X$ and $y$ in *Sequence* or *Graph* format.
**2.** Design $E_\theta$ {e.g. dual, perturbed, bidirectional, graph-based, etc} // Sec 3
**3.** Select training loss to learn $E_\theta$ and obtain $\theta^*$ // Sec 4
**Return** $\theta^*$

**[Test Phase]: Inference** // Sec 5
**Input**: $\theta^*$, $y^{\text{test}}$, Proposal $P$. // Sec 5
**4.** Obtain a list of $X$ candidates by $P$.
$L^{test} \leftarrow P(y^{test})$
**5.** $X^* = \arg\min_{X \in L^{\text{test}}} E_{\theta^*}(X, y^{\text{test}})$
**Return**: $X^*$

the forward prediction probability for reaction outcome prediction $p_\theta(y|X)$ can be written as $\frac{\exp(-E_\theta(X, y))}{\sum_{y'} \exp(-E_\theta(X, y'))}$ with the same form of energy function. Overall, the proposed framework works as follows: (1) design and train an energy function $E_\theta$ (Sec 3 and Sec 4), and (2) use $E_\theta$ for inference in retrosynthesis (Sec 5). See Fig 2 and Algorithm 1. Based on how to parameterize reactant and product molecule $X$ and $y$, the model designs can be divided into two categories: sequence-based and graph-based models.

## 3 MODEL DESIGN

### 3.1 SEQUENCE-BASED MODELS

Here we describe several sequence-based parametriztion to instantiate our EBM framework, which use SMILES string as representations of molecules. We first define the sequence-based notations.

Given a reactant molecule $x$, we denote its SMILES representation as $s(x)$. Superscript $s(x)^{(i)}$ denotes the character at $i$-th position of the SMILES string. For simplicity, we use $x^{(i)}$ when possible. Reactants of a chemical reaction are usually a collection of molecules: $X = \{x_1, x_2, .., x_j, .., x_{|X|}\}$, where $x_j$ is the $j$-th reactant molecule. The SMILES representation of a molecule set $X$, denoted as $s(X)$, is a concatenation of $s(x)$ for every $x$ in $X$ with "." in between: "$s(x_1).s(x_2)...s(x_{|X|})$". For simplicity, we use $X^{(i)}$ as the short form of $s(X)^{(i)}$ to denote the $i$-th position of this concatenated SMILES.

### 3.1.1 FULL ENERGY-BASED MODEL

We start by proposing a most flexible EBM that imposes the minimum restrictions on design of $E_\theta$. All the variants proposed in Sec 3.1 are special instantiations of this model (e.g. by specifying different $E_\theta$). The EBM is defined as follows:

$$p(X|y) = \frac{\exp\left(-E_\theta(X, y)\right)}{\sum_{X' \in \mathscr{P}(M)} \exp\left(-E_\theta(X', y)\right)} \propto \exp(-E_\theta(X, y)) \tag{2}$$

Here the energy function $E_\theta : \mathscr{P}(M) \times M \mapsto \mathbb{R}$ takes a molecule set and a molecule as input, and outputs a scalar value. $M$ defines the set of all possible molecules. $\mathscr{P}(\cdot)$ represents the power set. $\mathscr{P}(M)$ denotes domain of reactant sets $X$. Due to the intractability of the partition function, training involves additional information e.g., template or approximation of the partition (See Sec 4).

### 3.1.2 ORDERED MODEL

One design of energy function is factoring the input sequence in an autoregressive manner (Sutskever et al., 2014; Schwaller et al., 2019; Segler et al., 2018a).

$$p_\theta(X|y) = \exp\left(\sum_{i=1}^{|s(X)|} \log p_\theta(X^{(i)}|X^{(1:i-1)}, y)\right) = \exp\left(\sum_{i=1}^{|s(X)|} \log \frac{\exp\left(h_\theta(X^{(1:i-1)}, y)^\top e(X^{(i)})\right)}{\sum_{c \in S} \exp\left(h_\theta(X^{(1:i-1)}, y)^\top e(c)\right)}\right) \tag{3}$$

where $p_\theta(X^{(i)}|X^{(1:i-1)}, y)$ is parameterized by a transformer $h_\theta(p, q) : S^{|p|} \times S^{|q|} \mapsto \mathbb{R}^{|S|}$ where $|S|$ is vocabulary size. $e(c)$ is a one-hot vector with dimension $c$ set to 1. This choice of $h_\theta(p, q)$ enables efficient computing of the partition function, as it outputs a vector with length equal to $|S|$ to represent logits (unnormalized log probability) for each value in vocabulary. maximum likelihood estimator (MLE) is feasible for training, as this factorization allows tractable partition function.

### 3.1.3 DUAL MODEL

A different design is to leverage on duality of retrosynthesis and reaction prediction. They are a pair of mutual reversible processes that factorize the joint distribution in different orders, where reaction prediction is "forward direction" – $p(y|X)$) and retrosynthesis is the "backward direction" – $p(X|y)$. With additional prior modeling, the joint probability $p(X, y)$ factorizes to either $p(X|y)p(y)$ or $p(y|X)p(X)$. We propose a training framework that leverages on the duality of the forward and backward directions and performs consistent training between the two to bridge the divergence. The advantage of incorporating forward direction $p(y|X)$ has been showed (Guo et al., 2020), where the authors use a sequential Monte Carlo tree algorithm to search for reactants that agree with forward prediction score.

---

**Algorithm 2** Dual Model

**[Train Phase]: Learning**:
**Input:** Reactants X and product y.
Let $\theta = \{\gamma, \alpha, \eta\}$
Define $E_\theta$ as Eq (4)
$E_\theta = \log p_\gamma(X) + \log p_\alpha(y|X) + \log p_\eta(X|y)$
**1. Train backward**:
$\eta^* = \arg\min_\eta L_{dual} = \arg\max_\eta \widehat{\mathbb{E}}[\log p_\eta(X|y)]$
**2. Train prior and forward**: Plug in $\eta^*$
$p^{mix}(X, y) = \frac{1}{1+\beta}\hat{p}(X, y) + \frac{\beta}{1+\beta}\hat{p}(y)p_{\eta^*}(X|y)$
$\gamma^*, \alpha^* = \arg\min_{\gamma, \alpha} L_{dual}$
$\qquad = \widehat{\mathbb{E}}_{p^{mix}_{(X,y)}}[\log p_\gamma(X) + \log p_\alpha(y|X)]$
**[Test Phase]: Inference**:
**Input**: $\theta^* = \{\gamma^*, \alpha^*, \eta^*\}$, $y^{test}$, Proposal P.
$L \leftarrow P(y^{test})$
$X^* = \arg\min_{X \in L} E_{\theta^*}(X, y^{test})$
**Return** $X^*$

---

The advantage of the duality of reversible processes has been demonstrated in other applications as well. He et al. (2016) trained a reinforcement learning (policy gradient) model to achieve duality in natural language processing and improved performances. Wei et al. (2019) treated code summary and code generation as a pair of dual tasks, and improved efficacy by imposing symmetry between attention weights of LSTM encoder-decoder in forward and backward directions. Despite of their encouraging results, these models are not ideal for stable and efficient training for retrosynthesis, for example, policy gradient methods suffer from high variance. Therefore we propose a novel training method that is simple yet efficient for retrosynthesis task. We impose duality constraints by training forward direction on a mixture of samples drawn from the backward and original dataset. To our best knowledge, we are the first to apply duality to retrosynthesis and to impose duality constraints by samples drawn from one direction. The EBM is defined as:

$$p(X|y) \propto \exp\big(\log p_\gamma(X) + \log p_\alpha(y|X) + \log p_\eta(X|y)\big) = \exp(-E_\theta(X, y)) \quad (4)$$

where prior $p(X)$, forward likelihood $p(y|X)$, and backward posterior $P(X|y)$ are modeled as autoregressive models (Sec 3.1.2), parameterized by transformers with parameters $\gamma$, $\alpha$, and $\eta$. Note energy function can be designed free of choice.

Our consistent training is achieved by minimizing the "dual loss", where the duality constraints in the equation below are imposed to penalize KL divergence of the two directions, i.e., KL(backward|forward). For simplicity, we fix the backward probability in the dual loss, and therefore entropy $H(\text{backward})$ is dropped.

$$\gamma^*, \alpha^*, \eta^* = \arg\min_{\gamma,\alpha,\eta} \ell_{\text{dual}} \quad (5)$$

$$\ell_{\text{dual}} = -\Big( \underbrace{\widehat{\mathbb{E}}[\log p_\gamma(X) + \log p_\alpha(y|X)]}_{\text{forward direction}} + \underbrace{\beta\widehat{\mathbb{E}}_y\widehat{\mathbb{E}}_{X|y}[\log p_\gamma(X) + \log p_\alpha(y|X)]}_{\text{duality constraints}} + \underbrace{\widehat{\mathbb{E}}[\log p_\eta(X|y)]}_{\text{backward direction}} \Big)$$

$$(6)$$

$$= -\widehat{\mathbb{E}}_{p_{(X,y)}^{\text{mix}}}[\log p_\gamma(X) + \log p_\alpha(y|X)] - \widehat{\mathbb{E}}[\log p_\eta(X|y)] \quad (7)$$

where $\widehat{E}$ indicates expectation over empirical data distribution $\hat{p}(X, y)$. The duality constraints $\beta\widehat{E}_y\widehat{E}_{X|y}[\log p_\gamma(X) + \log p_\alpha(y|X)]$ is the expectation of the forward direction $\log p_\gamma(X) + \log p_\alpha(y|X)$ with respect to empirical backward data distribution $\hat{E}_y\hat{E}_{X|y}$, where $\hat{E}_y\hat{E}_{X|y}$ are approximated by samples drawn from $p_\eta(X|y)$, as $y$ is given so $p(y) = 1$ and $\beta$ is scaled parameter. In our implementation we use size $k$-beam search to draw samples. Combining "forward" and "duality constraints" terms (Eq 7), we can see that the first term of the dual loss is to train the forward direction on the mixture distribution of the original data and samples drawn from backward directions $p^{\text{mix}}(X, y) = \frac{1}{1+\beta}\hat{p}(X, y) + \frac{\beta}{1+\beta}\hat{p}(y)p_\eta(X|y)$. Put every piece together (Algorithm 2 and Fig 3 in Appendix). Here is our training procedure. Since we parameterize the three probabilities separately, the optimization of dual loss breaks into two steps:

- **Step 1: Train backward**. $\eta$ does not depend on forward direction under empirical data distribution. $\eta^* = \arg\min_\eta L_{\text{dual}} = \arg\max_\eta \widehat{E}[\log p_\eta(X|y)]$. $\eta$ can be learned by MLE.

- **Step 2: Train prior and forward**. We plug $\eta^*$ into $p_{\eta^*}^{\text{mix}}(X, y)$. $\gamma^*, \alpha^* = \arg\min_{\gamma,\alpha} L_{\text{dual}} = \arg\max_{\gamma,\alpha} \widehat{\mathbb{E}}_{p_{\eta^*(X,y)}^{\text{mix}}}[\log p_\gamma(X) + \log p_\alpha(y|X)]$. $\gamma, \alpha$ can be learned by MLE. Please also see ablation study of each components of dual loss in Appendix Sec A.6.

### 3.1.4   PERTURBED MODEL

In contrast to the ordered model that factorizes the sequence in one direction, we use a perturbed sequential model to achieve stochastic bidirectional factorization adapted from XLNet (Yang et al., 2019). In particular, this model permutes the factorization order (while maintaining position encoding of the original order) that is used in the forward autoregressive model.

$$p(X|y, z) = p(X^{(z_1)}, X^{(z_2)}, \ldots, X^{(z_{|s(X)|})}|y) = \prod_{i=1}^{|s(X)|} p_\theta(X^{(z_i)}|X^{(z_1:z_{i-1})}, y) \quad (8)$$

where the permutation order $z$ is a permutation of the original order sequence $z_o = [1, 2, \ldots, |X|]$ and $z_i$ denotes the $i$-th element of permutation $z$. Here $z$ is treated as hidden variable.

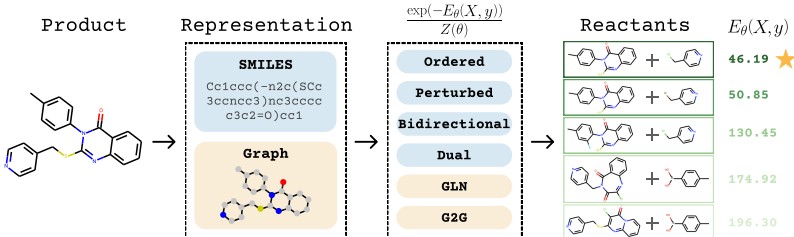

Figure 2: **EBM framework for retrosynthesis**. Given the product as input, the EBM framework (1) represents it as SMILES or a graph, (2) designs and trains the energy function $E_\theta$, (3) ranks reactant candidates with the trained energy score $E_{\theta^*}$, and (4) identifies the top $K$ reactant candidates. The best candidate has the lowest energy score (denoted by a star). The list of reactant candidates is obtained via templates or directly from the trained model.

During training, permutation order $z$ is randomly sampled and uses the following training objective:

$$p(X|y) \approx \exp\left(\mathbb{E}_{z \sim Z_{|s(x)|}}\left[\sum_{i=1}^{|X|} \log p_\theta(X^{(z_i)}|z_i, X^{(z_1:z_{i-1})}, y)\right]\right) \tag{9}$$

and the corresponding parameterization:

$$p_\theta(X^{(z_i)}|z_i, X^{(z_1:z_{i-1})}, y) = \log \frac{\exp\left(h(X^{(z_1:z_{i-1})}, z_i, y)^\top e(X^{z_i})\right)}{\sum_{c \in S} \exp\left(h(X^{(z_1:z_{i-1})}, z_i, y)^\top e(c)\right)} \tag{10}$$

where $z_i$ encodes which position index in the permutation order to predict next, implemented by a second position attention (in addition to the primary context attention). Note that Eq (9) is actually a lower bound of the latent variable model, due to Jensen's inequality. However, we focus on this model design for simplicity of permuting order in training. The lower-bound approximation is tractable for training.

### 3.1.5 BIDIRECTIONAL MODEL

An alternative way to achieve bidirectional context conditioning is the denoising auto-encoding model. We adapt bidirectional model from BERT (Devlin et al., 2018) to our application. The conditional probability $p(X|y)$ is factorized into product of conditional distributions of one single random variable conditioning on the others,

$$p(X|y) \approx \exp(\sum_{i=1}^{|s(X)|} \log p_\theta(X^{(i)}|X^{\neg i}, y)) \tag{11}$$

As presented in Wang & Cho (2019), although the model is similar to MRF (Kindermann, 1980), the marginal of each dimension in Eq (11) does not have a simple form as in BERT training objective. It may result in a mismatch between the model and the learning objective. This model can be trained by pseudo-likelihood (Sec 4.2)

### 3.2 GRAPH-BASED MODEL

Compared with the sequence-based model, the graph-based methods present chemical molecules, with vertices as atoms and edges as chemical bonds. This natural parameterization allows straightforward application of chemistry knowledge by sub-graph matching with templates or reaction centers. We instantiated three representative gragh-based approaches, namely NeuralSym (Segler & Waller, 2017b), GLN (Dai et al., 2019) and G2G (Shi et al., 2020), from the framework. Firstly, we introduce an important concept *template*, which can assist modeling, learning, and inference.

**Templates** are reaction rules extracted from existing reactions. They are formed by reaction centers (a set of atoms changed, e.g. to form or break bonds). A template $T$ consists of a product-subgraph pattern ($t_y$) and reactants-subgraph pattern(s) ($t_X$), denoted as $T := t_y \to t_X$, where $X$ is a molecular set. We overload the notation to define a *template operator* $T(\cdot) : M \mapsto \mathscr{P}(M)$ which takes a product as input, and returns a set of candidate reactant sets. $T(\cdot)$ works as follows: enumerate all the templates with product-subgraph $t_y$ matching with the given product $y$ and define $S(y) = \{T : t_y \in y, \forall T \in \mathcal{T}\}$, where $\mathcal{T}$ are available templates; then reconstruct the reactant candidates by instantiating reactant-subgraphs of the matched templates $R = \{X : t_X \in X, \forall T \in S(y)\}$. The output of $T(\cdot)$ is $R$. $T(\cdot)$ can be implemented by chemistry toolbox RDKit (Landrum, 2016).

### 3.2.1 Template prediction: NeuralSym

NeuralSym is a template-based method, which treats the template prediction as multi-class classification. The corresponding probability model under the EBM framework can be written as:

$$p(X|y) \propto \sum_{T \in \mathcal{T}} \exp(e_T^\top f(y)) \mathbb{I}\left[X \in T(y)\right] \tag{12}$$

where $f(\cdot)$ is a neural network that embeds molecule graph $y$, and $e_T$ is the embedding of template $T$. Learning such model requires only optimizing the cross entropy, despite that the number of potential templates could be very large.

### 3.2.2 Graph-matching with template: GLN

Dai et al. (2019) proposed a method of graph matching the reactants and products with their corresponding components in the template to model the reactants and template jointly, with the model:

$$p(X, T|y) \propto \exp(w_1(T, y) + w_2(X, T, y)) \cdot \phi_y(T) \phi_{y,T}(X) \tag{13}$$

where $w_1$ and $w_2$ are graph matching score functions, and the $\phi(\cdot)$ operators defines the hard template matching results. This model assigns zero probability to the reactions that do not match with the template. $p(X|y)$ can be obtained by marginalizing over all templates.

### 3.2.3 Graph matching with reaction centers, G2G

In contrast with GLN, Shi et al. (2020) proposed a method to predict reaction center directly. This method closely imitates chemistry experts when performing retrosynthesis: first identify reaction centers (i.e. where the bond breaks, denoted as $c$), then reconstruct $X$.

$$p(X|y) \propto \exp\left(\log\left(\sum_{c \in y} p(X|c, y)p(c|y)\right)\right) \tag{14}$$

All three methods mentioned above require the additional atom-mapping as supervision during training, while NeuralSym and GLN require template information during inference. So NeuralSym and GLN are template-based methods. Since atom mapping plus reaction centers have almost same information as templates, we denote G2G method as semi-template-based approach.

## 4 Learning

Training EBMs is to learn parameters $\theta$. In particular, we introduce three ways to learn exact (if applicable) or approximate maximum likelihood estimation (MLE) for full energy-based model (Sec 3.1.1), as this model includes other sequence-based EBM variants (ordered, perturbed, bidirectional, etc) by instantiating $E_\theta$ accordingly. Training EBMs with MLE is non-trivial because the partition function $Z(\theta)$ in Eq (1) is generally intractable. Computing $Z(\theta)$ involves approximation or additional information.

### 4.1 Approximate MLE: integration using template.

We use additional chemistry information: Templates. Direct MLE is not feasible because the partition function of Eq (2) involves enumerating full molecular set $M$, which is intractable. Here we use templates to get a finite support of the partition function. Specifically, we use template operator to extract a set of reactant candidates associated with $y$, denoted as $T(y)$ As the size of $T(y)$ is about tens to hundreds (not computationally prohibitive), we can perform exact inference of Eq (2) to obtain the MLE. We denote this training scheme as template learning.

### 4.2 Approximate MLE: pseudo-likelihood.

Alternatively, we can provide an approximation of Eq (2) via pseudo-likelihood (Besag, 1975) to enable training. Pseudo-likelihood factorizes the joint distribution into the product of conditional probabilities of each variable given the rest. Theoretically, the pseudo-likelihood estimator yields an exact solution if the data is generated by a model $p(X|y)$ and number of data points $n \to \infty$ (i.e., it is consistent) (Besag, 1975). For the full model, training is performed as:

$$p(X|y) \approx \exp\left(\sum_{i=1}^{|s(X)|} \log p_\theta(X^{(i)}|X^{\neg i}, y)\right) = \exp\left(\sum_{i}^{|s(X)|} \log \frac{\exp\left(g_\theta(X, y)\right)}{\sum_{c \in S} \exp\left(g_\theta(X', y; X'^{\neg i} = X^{\neg i}, X'^{(i)} = c)\right)}\right) \tag{15}$$

where the superscript $\neg$ indicates sequence except the $i$-th token and $g_\theta(p, q) : S^{|p|} \times S^{|q|} \mapsto \mathbb{R}$ is a transformer architecture that maps two sequences to a scalar. As bidirectional model Sec 3.1.5 and training approaches Sec 4.2 (approximate joint probability) factorizes in the same way, pseudo-likelihood is a convenient way to train this model.

### 4.3 EXACT MLE: TRACTABLE FACTORIZATION.

This training procedure only works for a special case of the full model, which has a tractable factorization of the joint probability, e.g., autoregressive models in ordered model (Sec 3.1.2) and perturbed models (Sec 3.1.4).

## 5 INFERENCE

With the trained $E_{\theta^*}$, inference identifies the best $X$ that minimizes the energy function for given $y^{\text{test}}$, i.e. $X^{\text{test}} = \arg\min_{X \in \mathcal{X}} E_{\theta^*}(X, y^{\text{test}})$. Directly solving the above minimization is again intractable, but the energy function can generally be used for ranking. Let $R$ denote the rank of candidate $X_i$ for the given $y^{\text{test}}$ (lower is better).

$$\{R(X_1) < R(X_2) \iff E_{\theta^*}(X_1, y^{\text{test}}) < E_{\theta^*}(X_2, y^{\text{test}})\} \tag{16}$$

Practically, as illustrated in Fig 2, one can use either template-based or template-free method to come up with initial proposals for ranking, as follows.

**Template-based Ranking (TB).** Templates can be used to extract a list of proposed reactant candidates by using templates. We use template operator $T(\cdot)$ (defined in Sec 3.2) to propose a list of candidate reactant sets from the input product $y$. **Template-free Ranking (TF).** In this paper, *template-free ranking* makes proposals using the learned structure prediction model. We use a simple autoregressive form for $p(X|y)$, which can draw the top $K$ most likely samples from this distribution using beam search, which is computational efficient.

## 6 EXPERIMENTS

**Experiment setup**: Dataset and evaluation used follow existing work (Coley et al., 2017b; Dai et al., 2019; Liu et al., 2017; Shi et al., 2020). We mainly evaluate our method on a benchmark dataset named USPTO-50k, which includes 50k reactions falling into ten reaction types from the US patent literature. we split the datasets into train/validation/test with percentage of $80\%/10\%/10\%$. Our evaluation metric is the top-$k$ exact match accuracy, referring to the percentage of examples where the ground truth reactant set was found within the top $k$ predictions made by the model. Following the common practice, we use RDKit (Landrum, 2016) to canonicalize the SMILES string from different representations in different methods. We augment USPTO with random SMILES as follows: (1) Replace each molecule in reactant set or product using random SMILES; (2) Random permute the order of reactant molecules.

We first present the evaluation of our best EBM variant (dual model) against existing methods (Appendix A.2) for both template-based and template-free approaches in Sec 6.1. Table 1 are our main results. Dual model statistics are extracted from Table 2 and 3. Then we provide comprehensive study on different variants of sequence-based EBMs in Sec 6.2. Table 2 serves as an ablation study to understand performance of different models. Then we provide template-free results in Table 3, which are our second main results. We provide time and complexity analysis in Appendix A.7.

### 6.1 COMPARISON AGAINST THE STATE-OF-THE-ART

Table 1 presents the main results. All the baseline results are copied from existing works as we share the same experiment protocol. The dual model is trained with randomized SMILES to inject order invariance information of molecule graph traversal. Note that other methods like graph-based variants do not require such randomization as the graph representation is already order invariant. We can see that, regarding top 1 accuracy when reaction type is unknown or known, our proposed dual model outperforms the current state-of-the-art methods by $9.9\%$ and $6.7\%$ for template-free setting, and $2.7\%$ and $3.5\%$ for template-based setting. Note that Dual-TF has quite close top-1 accuracy as Dual-TB, which demonstrates the discriminative ability of the designed energy function. The Dual-TB has higher top 10 accuracy due to the higher coverage from templates compared to the proposal obtained by $p(X|y)$ model. This suggest that with better proposal during inference, we can further boost the current performance.

### 6.2 SEQUENCE-BASED VARIANT EVALUATION

In this section, we mainly compare different energy based sequence models described in Sec 3.1, with both template-based and template-free evaluation criteria.

Table 1: **Top K exact match accuracy** of existing methods

| Category | Model | Reaction type unknown | | | | Reaction type known | | | |
|---|---|---|---|---|---|---|---|---|---|
| | | top1 | top3 | top5 | top10 | top1 | top3 | top5 | top10 |
| TB | retrosim (Coley et al., 2017b) | 37.3 | 54.7 | 63.3 | 74.1 | 52.9 | 73.8 | 81.2 | 88.1 |
| | NeuralSym (Segler & Waller, 2017b) | 44.4 | 65.3 | 72.4 | 78.9 | 55.3 | 76.0 | 81.4 | 85.1 |
| | GLN (Dai et al., 2019) | 52.5 | 69.0 | 75.6 | 83.7 | 64.2 | 79.1 | 85.2 | 90.0 |
| | Dual-TB (Ours) | **55.2** | **74.6** | **80.5** | **86.9** | **67.7** | **84.8** | **88.9** | **92.0** |
| Semi-TB | G2Gs (Shi et al., 2020) | 48.9 | 67.6 | 72.5 | 75.5 | 61.0 | 81.3 | **86.0** | **88.7** |
| | RetroXpert(Yan et al., 2020) | **65.6** | 78.7 | 80.8 | 83.3 | **70.4** | **83.4** | 85.3 | 86.8 |
| | GraphRETRO (Somnath et al., 2020) | 64.2 | **80.5** | **84.1** | **85.9** | 67.8 | 82.7 | 85.3 | 87.0 |
| TF | LSTM (Liu et al., 2017) | - | - | - | - | 37.4 | 52.4 | 57.0 | 61.7 |
| | Transformer (Zheng et al., 2019) | 43.7 | 60.0 | 65.2 | 68.7 | 59.0 | 74.8 | 78.1 | 81.1 |
| | Dual-TF (Ours) | **53.6** | **70.7** | **74.6** | **77.0** | **65.7** | **81.9** | **84.7** | **85.9** |

*Dual-TB/TF: Dual model with template-based or -free ranking.
Description of existing methods see Appendix A.2

Table 2: **Template-Based Proposal:** Top K accuracy of sequence variants

| Dataset | Models | Reaction type unknown | | | | Reaction type known | | | |
|---|---|---|---|---|---|---|---|---|---|
| | | Top 1 | Top 3 | Top 5 | Top 10 | Top 1 | Top 3 | Top 5 | Top 10 |
| USPTO 50k | Full model | 39.5 | 63.5 | 73.0 | 83.8 | 55.0 | 79.9 | 86.3 | **92.0** |
| | Ordered | 47.0 | 67.4 | 75.4 | 83.1 | 60.9 | 80.9 | 85.8 | 90.2 |
| | Perturbed | 42.9 | 58.7 | 63.9 | 69.6 | 56.6 | 73.6 | 77.2 | 81.6 |
| | Bidirectional | 16.9 | 34.4 | 45.6 | 61.1 | 31.4 | 57.0 | 69.8 | 81.3 |
| | Dual | **48.4** | **69.1** | **77.0** | **84.4** | **61.7** | **81.5** | **86.9** | 91.1 |
| Augmented USPTO 50k | Ordered | 54.2 | 72.0 | 77.7 | 84.2 | 66.4 | 82.9 | 87.4 | 91.0 |
| | Perturbed | 47.3 | 64.6 | 70.4 | 75.8 | 64.2 | 79.8 | 83.3 | 86.4 |
| | Bidirectional | 23.5 | 43.7 | 54.3 | 69.5 | 41.9 | 66.3 | 75.6 | 84.6 |
| | Dual | **55.2** | **74.6** | **80.5** | **86.9** | **67.7** | **84.8** | **88.9** | **92.0** |

### 6.2.1 TEMPLATE-BASED RANKING

Table 2 provides the results of template-based ranking (Sec 5 TB) for each sequence model variant described in Sec 3.1. Random SMILES has been shown useful for RNN and LSTM models Arús-Pous et al. (2019). We want to convey the message to the community that augmentation using random SMILES is critical for ensuring best performance for transformer by preventing overfitting. Without reiterating good performance for the dual variant, we focus on discussion of variants with undesired performance. The perturbed sequential model (Sec 3.1.4) and bidirectional model (Sec 3.1.5) are inferior to dual or ordered models, where the main reason possibly comes from the fact that the learning objective approximates the actual model in Eq (9) and Eq (11) poorly, and thus leads to discrepancy between training and inference (See more discussion in Appendix). The full model (Sec 3.1.1) despite being most flexible and achieving best top 10 performance when type is given, would suffer from high computation cost due to the explicit integration even with the templates. In addition to the understanding of individual models throughout the comprehensive study, we find it is important to balance the trade-off between model capacity and learning tractability. A powerful model without effective training would be even inferior to some well trained simple models. Our dual model makes a good balance between capacity and learning tractability. Please see discussion about semi-template based methods in Appendix A.2, A.3, A.4.

### 6.2.2 TEMPLATE-FREE RANKING

Table 3 presents the results for template-free evaluation (Sec 5 TF). Template-free evaluation approach proposed in this paper requires a proposal model with good coverage and a ranking model with good accuracy. We explored various combinations of proposal-ranking pairs. The proposal model evaluated is the ordered model trained on USPTO50K and augmented USPTO50K, respectively. The ranking model is the dual model trained on augmented data, as it performs the best in Table 2. Our best performer is ordered-proposal (USPTO 50K)-dual-ranking (aug USPTO 50K) model. We reached top 1 accuracy as 53.6% and top 10 accuracy as 77.0% when type is unknown. Note this accuracy are already close to template-based state-of-the-art. A case study showing how dual model improves accuracy upon proposal is given Fig 4 in Appendix, where it shows how the energy based re-ranking refines the initial proposal. One interesting observation is that, the proposal ordered model trained on augmented data has higher top 1 accuracy but much lower top 10 accuracy. This indicates that such proposal has quite low coverage in the prediction space. We observed that the model learned on augmented dataset learns various representations of the same molecule (due to usage of random SMILES). A certain percentage of proposed candidates are the same after canonicalization, which is good for top 1 prediction but undesired for proposal.

Table 3: **Template-free**: Translation Proposal and Dual Ranking

| Type | Proposal | | | | | | Re-rank | | | | |
|---|---|---|---|---|---|---|---|---|---|---|---|
| | Proposal model | Top 1 | Top 5 | Top 10 | Top 50 | Top 100 | Rank model | Top 1 | Top 3 | Top 5 | Top 10 |
| No | Ordered on UPSPTO | 44.4 | 64.9 | 69.9 | 77.2 | 78.0 | Dual trained on Aug USPTO | 53.6 | **70.7** | **74.6** | **77.0** |
| | Ordered on Aug USPTO | 53.2 | 54.7 | 55.6 | 60.5 | 60.5 | | **54.5** | 60.0 | 60.4 | 60.5 |
| | - | - | - | - | - | - | SOTA (SCROP (Zheng et al., 2019)) | 43.7 | 60.0 | 65.2 | 68.7 |
| Yes | Ordered on USPTO | 56.0 | 76.1 | 79.7 | 85.2 | 86.4 | Dual trained on Aug USPTO | 65.7 | **81.9** | **84.7** | **85.9** |
| | Ordered on Aug USPTO | 64.7 | 66.5 | 67.3 | 69.7 | 75.7 | | **66.2** | 75.1 | 75.6 | 75.7 |
| | - | - | - | - | - | - | SOTA (SCROP (Zheng et al., 2019)) | 59.0 | 74.8 | 78.1 | 81.1 |

## 7 CONCLUSION

In this paper we proposed an unified EBM framework that integrates multiple sequence- and graph-based variants for retrosynthesis. Assisted by a comprehensive assessment, we provide a critical understanding of different designs. Based on this, we proposed a novel variant—dual model, which outperforms state-of-the-art in both template-based and template-free setting.

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

# A APPENDIX

## A.1 TERMINOLOGY: REACTION CENTER AND SYNTHONS

Reaction center of a chemical reaction are the bonds that are broken or formed during a chemical reaction. For retrosynthesis, reaction centers are bonds exist in product, but do not exist in reactants. One chemical reaction may have multiple reaction centers. Synthons are the sub-parts extracted from the products by breaking the bonds in the reaction center. Synthons are usually not valid molecules with ∗ to indicate the broken ends in the reaction centers.

## A.2 EXISTING METHODS

We evaluate of our approach against several existing methods, including both template-based and template-free approaches. Specifically, for **Template-free ones**: `Transformer` (Zheng et al., 2019) is a transformer based approach that trains a second transformer to identify the wrong translations and remove them. `LSTM` (Liu et al., 2017) is a sequence to sequence approach that use LSTM as encoder and decoder. For **Template-based ones:** `retrosim.` (Coley et al., 2017b) selects template for target molecules using fingerprint based similarity measure between targets and templates; `neuralsym` (Segler & Waller, 2017b) performs selection of templates as a multiple-class problem using MLP; `GLN` builds a template induced graphical model and makes prediction with approximated MAP.

For **Semi-Template based ones**: The three methods: `G2Gs` (Shi et al., 2020), `GRetroXpert` (Yan et al., 2020), and `GraphRETRO` (Somnath et al., 2020), share the same idea: infer reaction center to generate synthons, and then complete the missing pieces (aka "leaving groups") in synthons to generate reactants. The three methods share the same first step – phrasing identification of reaction center bond as a classification problem. They have different second steps: from synthons to reactants. `G2G` uses graph generation via variational inference. `RetroXpert` use transformer. `GraphRETRO` predefine s list of possible leaving groups and phrase the problem as a classification problem. Please note `G2G` and `RetroXpert` are generative model that can be applied to unseen molecules, whereas `GraphRETRO` requires the pre-defined list, which simplies the problem significantly and may not apply to molecules with unseen leaving groups.

## A.3 TEMPLATE-FREE METHODS AND SEMI-TEMPLATE BASED METHODS

In this paper, we refer **template-free** methods as the approaches that do not require any chemistry hand-crafted features, including templates, sub-parts of templates, reaction centers, and atom mapping. More precisely, our experiment setup is **full automatic template-free methods**. We are interested in this set up because

- **These handcrafted features are often not accessible for real world application.** Getting these features ether requires intensive human efforts or non-trivial computational challenges. For example, the ground truth "reaction center" used in semi-template based methods, have to be manually labeled by human experts when atom-mapping is not available. So it is prohibitive for large datasets. Alternatively, reaction centers can be computationally extracted using atom-mapping. However, obtaining good quality atom mapping for large datasets itself is an equally challenging problem.

- **Think beyond existing rule (e.g. handcrafted features).** The future retrosynthesis models should be able to think beyond existing chemical rules, and achieve full automation without handcrafted features. We refer existing rules as templates, reaction center, atom mapping, reaction type, etc. Thinking beyond exiting rules denotes generating unknown chemical rules. Similar to the computer vision field (e.g. imagenet classification), people used handcrafted features at the beginning and all move to full automation without any handcrafted features because full automation achieves much better results. Unfortunately, we are not there yet. Including hand-crafted features often lead to a dramatic improvement (10%+) regardless of model design. For example, well-known examples are reaction type and templates. Please also so next section.

In this paper, we refer methods that use additional chemistry features during training as **semi-template-based** methods. For example, `G2Gs` (Shi et al., 2020), `GRetroXpert` (Yan et al., 2020), and `GraphRETRO` (Somnath et al., 2020). See description of the methods in Appendix A.2. These methods do not apply template-matching explicitly, so they are not template-based methods. However, they use "reaction centers" as additional information to supervise their algorithm during training. The reaction centers preserve almost equivalent information as templates – once reaction centers are given, the product can be broken into two parts (denoted as synthons). The templates can be recovered by removing nonessential atoms from synthons and product. In addition to reaction centers, `RetroXpert` also add additional features "the product side of template" to its atom features. `GraphRETRO` has a pre-defined list of leaving groups. Therefore, we denote G2G, RetroXpert and GraphRETRO as semi-template based methods.

### A.4    DUAL MODEL PERFORMANCE WITH REACTION CENTER GIVEN

Semi-template based methods `RetroXpert` (Yan et al., 2020), and `GraphRETRO` (Somnath et al., 2020) achieved impressive results. Their performance even outperforms template-based approach.

The models presented in this paper have a different experiment setup from `RetroXpert` and `GraphRETRO`. Our setup does not use additional chemical information: reaction center, during training, which makes a head-to-head comparison not meaningful. To further investigate the effect of this additional chemistry features, we asked what will happen if our models also have the information of "reaction center"? Can we have the same or better performance when reaction center is given? As a proof of concept, we incorporated the reaction center to the dual model as follows: we first generate synthons from product using the given reaction centers, and the then concatenate the generated synthons to the input of our dual model.

Table 4: **Template-free with reaction center**: Translation Proposal and Dual Ranking

| Type | Proposal | | | | | | Re-rank | | | | |
|---|---|---|---|---|---|---|---|---|---|---|---|
| | Proposal model | Top 1 | Top 5 | Top 10 | Top 50 | Top 100 | Rank model | Top 1 | Top 3 | Top 5 | Top 10 |
| No | Ordered on UPSPTO | 65.8 | 83.8 | 86.2 | 90.3 | 90.7 | Dual trained on Aug USPTO | **70.1** | **86.3** | **89.0** | **90.3** |
| | Ordered on Aug USPTO | 68.7 | 70.0 | 70.5 | 72.1 | 74.7 | | 69.9 | 74.4 | 74.6 | 74.7 |
| | - | - | - | - | - | - | RetroXpert (Yan et al., 2020) | 65.6 | 78.7 | 80.8 | 86.8 |
| Yes | Ordered on USPTO | 67.7 | 85.7 | 88.0 | 91.6 | 91.9 | Dual trained on Aug USPTO | **73.2** | **88.4** | **90.8** | **91.6** |
| | Ordered on Aug USPTO | 71.4 | 73.3 | 73.8 | 75.3 | 77.7 | | 73.1 | 77.4 | 77.7 | 77.7 |
| | - | - | - | - | - | - | RetroXpert (Yan et al., 2020) | 70.4 | 83.4 | 85.3 | 86.8 |

Compared with Table 3 (when no synthons information is available), Table 4 shows adding reaction centers can boost accuracy significantly: When reaction type is unknown, the accuracy of final retrosynthesis improves from 54.5% to 70.1%. When reaction type is known, the accuracy improves from 66.2% to 73.2%. An interesting observation is that when reaction centers are presented, the effect of reaction type and augmentation on positively increasing the accuracy is reduced. One explanation is reaction center is so important that it dominates all the other useful features or tricks.

Although in the above experiments, we are given the reaction center, that is we do not infer them, Table 4 still serves well as a proof of concept that incorporating useful handcrafted chemistry features can significantly boost accuracy regardless of model design. To be more precise, Table 4 is an upper bond of Dual model when we use inferred reaction center as input if we are given reaction center as additional information during training. Dual with reaction center given is NOT template-free methods. It is semi-template based methods as RetroXpert, GraphRETRO, G2G.

In addition to reaction centers, `RetroXpert` also add additional features "the product side of template" to its atom features. `GraphRETRO` has a pre-defined list of leaving groups. Incorporating those information (i.e. product side if template) in our setup may boost accuracy significantly again.

We want to avoid pursuing the path of chasing the state of the art by incorporating many useful handcrafted features, as it is contradictory to full automatic template-free methods. We want to deliver the message to the community that the performance of models should be compared under the same setup. The performance gain may be a result of using extra information, instead of modeling.

## A.5 DUAL MODEL FIGURE

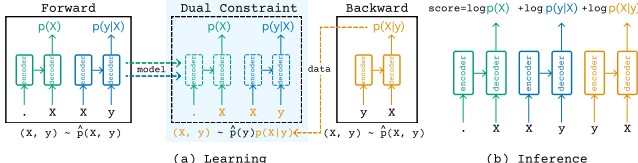

(a) Learning          (b) Inference

Figure 3: **Dual model. (a) Learning** consists of training three transformers: prior $p(X)$ (green), likelihood $p(y|X)$ (blue), and backward $p(X|y)$ (orange). Dual model penalizes the divergence between forward $p(X)p(y|X)$ and backward direction $p(y|X)$ with Dual constraint (highlighted). **(b) Inference** Given reactant candidates list, we rank them using Eq (4).

## A.6 ABLATION STUDY OF THE DUAL LOSS

We perform ablation study to investigate whether training with the dual loss, in particular the dual constraint, is the reason for the performance improvement. Recall the dual loss is given in Eq (6) and the dual constraint is the middle term in Eq (6). See Table 5. The "dual" row indicates training with dual loss and the results are taken from Table 1. $\widehat{\mathbb{E}}[\log p_\gamma(X) + \log p_\alpha(y|X) + \log p_\eta(X|y)]$ is dual loss without the dual constraint. $\widehat{\mathbb{E}}[\log p_\alpha(y|X) + \log p_\eta(X|y)]$ is without prior $\log p_\gamma(X)$. $\log p_\eta(X|y)]$ is only including backward direction. Table 5 shows each component of the dual loss contribute to the performance positively.

Table 5: **Ablation Study of dual loss when reaction type is known**

| Aug USPTO | Top 1 | Top 3 | Top 5 | Top 10 |
|---|---|---|---|---|
| Dual | **67.7** | **84.8** | **88.9** | **92.0** |
| $\widehat{\mathbb{E}}[\log p_\gamma(X) + \log p_\alpha(y|X) + \log p_\eta(X|y)]$ | 67.0 | 84.7 | 88.9 | 91.95 |
| $\widehat{\mathbb{E}}[\log p_\alpha(y|X) + \log p_\eta(X|y)]$ | 66.1 | 82.8 | 87.6 | 91.3 |
| $\widehat{\mathbb{E}}[\log p_\eta(X|y)]$ | 60.9 | 80.9 | 85.8 | 90.2 |

## A.7 TIME AND SPACE COMPLEXITY ANALYSIS

In this section, we provide time and space complexity regarding model design choices. As the main bottleneck is the computation of transformer model, we measure the complexity in the unit of transformer model calls. For all the models, the inference only requires the evaluation of (un-normalized) score function, thus the complexity is $O(1)$; For training, the methods that factored have an easy form of likelihood computation, where a diagonal mask is applied to input sequence so that autoregressive is done in parallel (not $|s(x)|$ times), so it requires $O(1)$ model calls. This include ordered/perturbed/bidirectional/dual models. For the full model trained with pseudo-likelihood, it requires $O(|X| \cdot |S|)$ calls due to the evaluation per each dimension and character in vocabulary. Things would be a bit better when trained with template-based method, in which it requires $O(|T(y)|)$ calls, which is proportional to the number of candidates after applying template operator.

As the memory bottleneck is also the transformer model, it has the same order of growth as time complexity with respect to sequence length and vocabulary size. In summary we can see the Full model has much higher cost for training, which might lead to inferior performance. Our dual model with a consistency training objective has the same order of complexity than other autoregressive ones, while yields higher capacity and thus better performance.

## A.8 TRANSFORMER ARCHITECTURE

The implementation of variants in framework is based on OpenNMT-py (Klein et al., 2017). Following (Vaswani et al., 2017), transformer is implemented as encoder and decoder, each has a 4

self-attention layers with 8 heads and a feed-forward layer of size 2048. We use model size and word embedding size as 256. Batch size contains 4096 tokens, which approximately contains 20-200 sequences depending on the length of sequence. We trained for 500K steps, where each update uses accumulative gradients of four batches. The optimization uses Adam (Kingma & Ba, 2014) optimizer with $\beta_1 = 0.9$ and $\beta_2 = 0.998$ with learning rate described in (Vaswani et al., 2017) using 8000 warm up steps. The training takes about 48 hours on a single NVIDIA Tesla V100. The setup is true for training transformer-based models, including ordered sequential model (Sec 3.1.2), perturbed sequential model (Sec 3.1.4), bidirectional model (Sec 3.1.5), dual model (Sec 3.1.3). As for full model (Sec 3.1.1), each sample contains 20-500 candidates. We implemented as follows: each batch only contains one sample. Its tens or hundreds of candidates are computed in parallel within the batch. The model parameters are updated when accumulating 100 batches to perform one step of update.

## A.9   EXAMPLE OF CASE STUDY

Here we provide another case study showing with dual model ranking (Sec 3.1.3), the accuracy improves upon translation proposal. Please see Fig 4 and Fig 5.

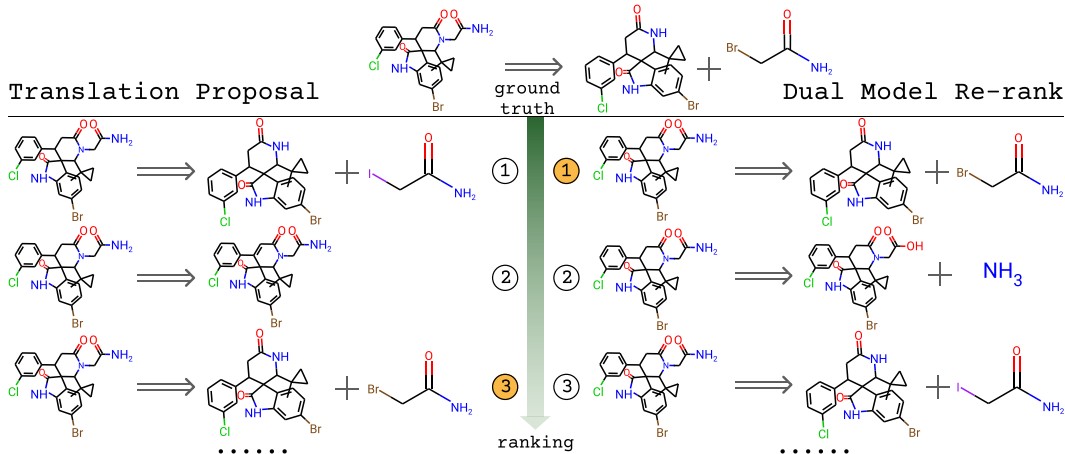

Figure 4: **Dual ranking improves upon translation proposal.** Left and right column are the top three candidates from translation proposal and dual re-ranking of the proposal. Ground truth (GT) is given at the top and is labeled orange in the middle. By dual re-ranking, the GT ranks the first place, whereas the 3rd place in the proposal. Note that the first place in the proposal is only one atom different from GT (Br vs I), indicating the dual model is able to identify small changes in structure.

## A.10   DISCUSSION

**V.1  Full model** (Sec 3.1.1) Full model (Sec 3.1.1) with template learning reaches accuracy of $39.5\%$ and $53.7\%$ on USPTO50k data-sets. Full model is partially limited by expensive computation due to the number of candidates per product.

**V.2  Perturbed sequential model** (Sec 3.1.4) Perturbed sequential model has about $\sim 4\%$ accuracy loss in top 1 accuracy compared with ordered model (Sec 3.1.2). We argue the reason are as follows: firstly, we designed $E_\theta$ as the middle term of Eq. Eq (9) to facilitate perturbing the order during training, following (Yang et al., 2019). However, due to Jensen's inequality, this design is not equal to $P(X|y)$, which causes discrepancy in ranking (inference).

**V.3  Bidirectional model** (Sec 3.1.5) Bidirectional model, however, does not perform well in our experiments. The bidirectional-awareness makes the prediction of one position given all the rest of the sequence $p(X^{(i)}|X^{\neg i}, y)$ almost perfect ($99.9\%$ accuracy in token-level). However, due to the gap between pseudo-likelihood and maximum likelihood, i.e., $\log P(X|y)$, the performance for predicting the whole sequence will be inferior, as we observed in the experiments.

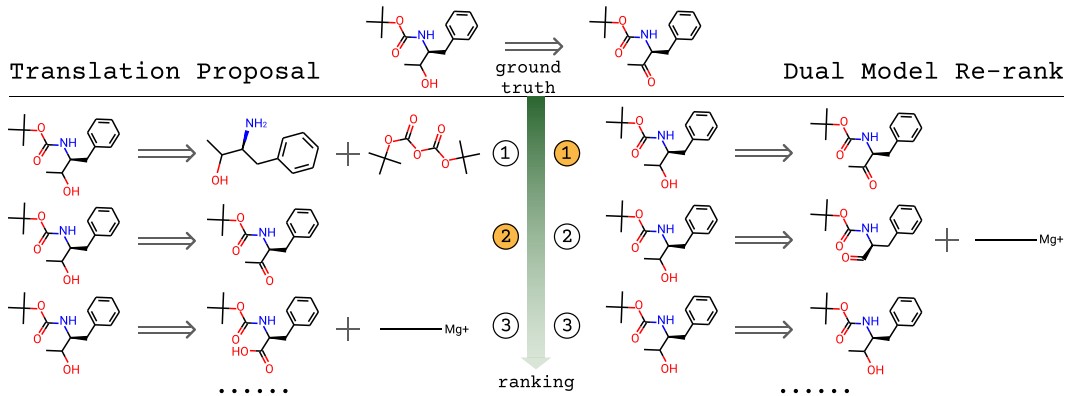

Figure 5: **Dual ranking improves upon translation proposal.** Another example. Descriptions see Fig 4

## A.11 ALTERNATIVE OF SMILES: DEEPSMILES AND SELFIES

In this section, we explore the effect of prepossessing procedure of sequence-based model, e.g. inline representation of molecular graph, in effecting performance of sequence-based model. In particular, deepSMILES (Dalke, 2018) and SELFIES (Krenn et al., 2019) are alternatives to SMILES. Without loss of fairness, we evaluated these representations using Ordered sequential model (Sec 3.1.2)The results indicate SMILES work the best. We speculate the reason are deepSMILES and SELFIES are on average longer than SMILES, leading to higher probability of making mistakes on token level and therefore low sequence-level accuracy.

Table 6: **deepSMILES and SELFIES**

| **SMILES** | | | | |
| --- | --- | --- | --- | --- |
| Models | Top 1 | Top 3 | Top 5 | Top 10 |
| Ordered | 47.0 | 67.4 | 75.4 | 83.1 |
| **deepSMILES** | | | | |
| Ordered | 46.08 | 65.87 | 73.54 | 81.51 |
| **Selfies** | | | | |
| Ordered | 43.00 | 62.51 | 70.16 | 79.07 |

## A.12 TRANSFORMER IMPLEMENTATION OF PERMUTATION INVARIANT OF REACTANT SET

Transformer has a position encoding to mark the different locations on an input sequence. We modified the position encoding such that each molecule starts with 0 encoding, instead of the concatenated position in the reactants sequence. The results are Table 7. We can see that this position encoding is beneficial for non-augment data, but not augment data, as the latter has already considered the permutation invariance order of reactants by data augmentation. In this paper, we use data augmentation to maintain order-invariant for reactants.

Table 7: **Transformer model with permutation invariant position encoding**

| Reaction type is unknown | USPTO 50k | | | | |
|---|---|---|---|---|---|
| Models | Top 1 | Top 2 | Top 3 | Top 5 | Top 10 |
| Ordered | 46.97 | 60.71 | 67.39 | 75.35 | 83.14 |
| Ordered + Permutation invariant | 47.29 | 61.29 | 68.08 | 75.37 | 83.36 |
| | Augmented data | | | | |
| Ordered | 54.24 | 66.33 | 72.02 | 77.67 | 84.22 |
| Ordered + Permutation invariant | 53.45 | 66.61 | 72.58 | 78.33 | 85.42 |

