# OpenReview forum: "Energy-based View of Retrosynthesis"
_ICLR.cc/2021/Conference — Reject_

### Official Review · AnonReviewer3 · 2020-10-27
**ICLR 2021 Conference Paper2550 AnonReviewer3**

**Rating:** 5
**Confidence:** 4

**Review:**

# Summary #
This paper introduces a re-interpretation of seq2seq and graph2graph retrosynthesis models with energy based models (EBMs).
EBM is a general log-linear framework to model joint distributions of a feature variable X and a target variable y, first introduced to connect discriminative (DNN) models with generative (DNN) models. This paper shows the reformulations of typical seq2seq retrosynthesis models and graph2graph models. Also, the paper proposes a dual training model that optimizes both the forward path model and the backward path model.
Experimental results show that the dual training models perform better than conventional retrosynthesis models in template-based and template-free retrosynthesis frameworks.

# Comments #

I have two major concerns.

First, I cannot clearly understand a new insight or knowledge that is brought by the EBM-based re-formulation of retrosynthesis models.
Any probabilistic models can be described by the full joint distribution of all variables. In my understanding, the EBM is a way of modeling the joints with the log-linear model plus the potential function.
It seems for me that the current manuscript is successful in re-wiring the existing retrosynthesis frameworks into EBMs, but that is all.
Please clarify what readers can learn about "connections and ... differences between models, ...understanding of model design [abstract]" from the EMB modelings.

Second, the State-of-the-Art of the retrosynthesis in the manuscript is somewhat outdated.
To the best of my knowledge, the current best-performing retrosynthesis models are [1] and [2], and the scores of these models are higher than the reported results of the proposed dual models.

I think the current manuscript needs modifications to appropriately cite these papers and replace the standpoint of the submitted work.

Personally, I am interested in how the EMB can interpret these latest models.


[1] Somnath+, "Learning graph models for template-free retrosynthesis", arXiv:2006.07038, June 2020.
[2] Yan+, "RetroXpert: Decompose retrosynthesis prediction like a chemist", chemrxiv:11869692.v3, June 2020.

# Evaluation points #

(+) First to apply the EBM for retrosynthesis prediction models

(+) Proposed a forward/backward simultaneous (dual) training

(--) It is unclear what we can learn by rewriting the retrosynthesis models with EBM.

(-) Reported results do not update the State-of-the-Art prediction accuracy of retrosynthesis models in the literature

---

> ### Author Response · Authors · 2020-11-24
> **Reply to R3:**
>
> Thank you for the insightful comments.  We have loaded the revised paper to address the comments, and the changes are highlighted in blue. Please see our response:
>
> ### Q1: Why is the EBM framework beneficial? What do we get from unifying models into EBM?
>
> Thank you for the question. EBMs provide a way to map high dimensional data (X, y) into an unnormalized probability distribution. EBM works on a wide range of problems and demonstrates good performance. The community has accumulated lots of knowledge of how to design energy scores and to train them efficiently. That prior knowledge can generalize to new problems or new model designs.
>
> EBM provides the following:
>
> **1. Principled ways for training**.
>
> EBMs have many off-the-shelf training methods that are ready to use, including but limited to contrastive divergence (Hinton, 2002), pseudo-likelihood (PL) (Besag, 1975), conditional composite likelihood (CL) (Lindsay, 1988), score matching (SM) (Hyv¨arinen, 2005), minimum (diffusion) Stein kernel discrepancy estimator (DSKD) (Barp et al.,2019), non-local contrastive objectives (NLCO) (Vickrey et al., 2010), minimum probability flow (MPF) (SohlDickstein et al., 2011), and noise-contrastive estimation (NCE) (Gutmann and Hyvarinen, 2010).
>
> For example, if we design the energy score function as the “bidirectional model” (eq11 in Sec 3.1.5),  with EBM in mind, it is natural to think of using pseudo-likelihood (PL) as the training approach.
>
> Additionally, there are different ways to design loss see Sec 2.2 of
> LeCun et al tutorial http://yann.lecun.com/exdb/publis/pdf/lecun-06.pdf page 6
>
> **2. Better inference**
>
> EBM model’s inference is flexible. LeCun et al tutorial Page 6 (inference).
>
> EBM can not only make inferences such as “classification” or “prediction”, e.g. what is best y (label) for X (input image), but also EBM’s inference can be used to rank possible (X, y)-pairs. From the EBM point of view, we can decouple the model, learning (training), and inference (testing). For example, one can use the autoregressive model for modeling and learning, while during inference we can use it as an energy function to get better samples (via re-ranking as we used, or doing Gibbs-sampling to refine the samples).
>
> **3. Advance understanding of different model designs.**
>
> * Extract commonalities and differences between EBM variants.
>
> One example, RetroXpert, GraphRETRO, and G2G share the same energy score format (eq 14) but the differences are the various instantionans of $p(X|y, c)$. Another example is comparing the energy score design ordered model (eq3), perturbed model (eq8), and bidirectional model (eq11).  We see their bidirectional degree: ordered < perturbed <  bidirectional
> The experiments show although bidirectional provides more information to the model, at the same time adds challenges in modeling $p(X|y)$ (from $p(X_i| X_{\setminus i}, y)$ to $p(X|y)$ ) and training.  In addition, some variants share the same training procedure  (ordered model and perturbed model both use MLE).
>
> * Understand the strengths and limitations in model design.
>
> Full model versus ordered model.  The full model is more flexible that do not impose restrictions on model design, however, the expressional power adds challenges in training, say enumerating all possible molecules.
> In contrast, the ordered model is restricted to the autoregressive model, but the training is much tractable (MLE)
>
> * Compare the complexity of learning or inference.
>
> See appendix A.7 "TIME AND SPACE COMPLEXITY ANALYSIS"
>
>
> ### Q2: Not SOTA, compared with RetroXpert (Yan et al,  NeurIPS 2020) and GraphRETRO (Somnath et al, 2020).
>
> Please kindly refer to the response to all reviewers with the title "Reply to R1, R2, R3, R4: Not SOTA...”.
>
> We have a different setup. These RetroXpert and GraphRETRO are not template free. The lower results are due to that we are not using extra chemical features. If we add those features to the dual model, we get better results than RetroXpert.  Dual: 70.1%; 73.2% RetroXpert: 65.6%; 70.4%  (Table 4 in Appendix A. 4 of our revised paper).
>
>
> ### Q3: “I am interested in how the EMB can interpret these latest models.” How to interpret the latest model such as RetroXpert and GraphRETRO into the EBM framework?
> Please also see Appendix A.1 and A2 of revised paper.
>
> Yes,  RetroXpert and GraphRETRO can be integrated into our EBM framework in a way that is similar to G2G (Sec 3.2.3 in our paper. eq14). The difference between the three is parameterizing $p(X|c, y)$, which is the step that generates reactants from synthons.  Synthons are subgraphs extracted from the products by breaking the bonds in the reaction centers.  Reactants are generated by completing the “missing pieces” (“leaving groups” in GraphRETRO) in synthon. RetroXpert uses transformers to complete the missing pieces of reactant from synthons; GraphRETRO uses a predefined list of leaving groups and selects the missing pieces; G2G uses graph generation.

---

### Official Review · AnonReviewer1 · 2020-10-27
**Interesting analysis of the problem of retrosynthesis using EBMs**

**Rating:** 5
**Confidence:** 5

**Review:**

In this paper, the authors use the framework of energy based models to describe several known approaches for ML-based retrosynthesis in a unified way. This allows to combine retrosynthetic (backward) and reaction prediction (forward) in a principled way.
Based on this analysis, a dual model is proposed which used a duality constraint as a regulariser, leading to improved performance over several baselines models, but not over SOTA (see below!).

The theoretical analysis alone is very interesting, and well described. However, I have a few concerns with missing ablation experiments and the positioning, which could be addressed to make the paper stronger in my opinion.

Overall, I think this paper should be accepted at ICLR after the following points have been addressed (or the authors commit to provide the additional data in the camera ready version if the time of the rebuttal phase is too short to run additional experiments). However, in the current form, the paper is not ready. My evaluation is for the current form of the paper, and I am happy to change it significantly during the rebuttal.


Ablation Experiments:

Is the duality constraint actually needed? It would be important to perform the ablation experiment where $\beta$ is set to 0.
In other words, what happens if you just use $\log p(X|y) + \log  p(y|X)$ to rank the candidates? If the authors perform these additional experiments and report the numbers, I will increase my score regardless of the outcome of the experiments.


Prior work:

The state of the art claim is not correct. Yan et al achieve higher performance on the same dataset https://chemrxiv.org/articles/preprint/Interpretable_Retrosynthesis_Prediction_in_Two_Steps/11869692/2 which has already been published last February!
Therefore, please remove the SOTA claim from the paper, and acknowledge the Yan et al work.
For this reviewer well-motivated modelling, honest analysis and proper experimentation is more important than chasing SOTA, and not achieving SOTA will not affect the evaluation negatively.

Also, the connection of forward reaction and retrosynthesis prediction via Bayes Theorem has been studied here https://arxiv.org/abs/2003.03190 which should be acknowledged.

Other work preceding has also used a combination of backward and forward prediction. For example, Segler et al Nature 2018 and Coley et al Science 2019 use a model for p(X|y) to propose disconnections combined with a model for p(X,y) to remove low probability solution.

The usefulness of SMILES augmentation has been previously shown, please cite the previous work by Arus Pous et al https://jcheminf.biomedcentral.com/articles/10.1186/s13321-019-0393-0

The Coley et al 2017a paper that the authors cite in the introduction is a wonderful paper, however, it is not concerned with retrosynthesis. I would suggest to cite the review article by Strieth-Kalthoff et al https://doi.org/10.1039/C9CS00786E instead, which provides a good overview over ML approaches for (retro)synthesis. Furthermore, I would suggest to cite Segler & Waller 2017 in the introduction, which was the first paper to suggest deep neural networks for both retrosynthesis & reaction prediction (see also https://doi.org/10.1016/j.ddtec.2020.06.002 )
Coley 2017a should be cited later in the context of forward prediction.

Comments:

Table 3 is a bit unclear. With Type, do you mean the reaction type is given?

small things:
- in eq 13, the is a low dot, should this have been a centred dot for multiplication?

- Please don't use Google house fonts for your figures,  to maintain anonymity.

---

> ### Author Response · Authors · 2020-11-24
> **Reply to R1**
>
> Thank you for the insightful comments and helpful suggestions! We have loaded a revised paper to address the comments, and the changes are highlighted in blue. Please see our response:
>
> **Q1: Ablation Experiments on with or without dual constraint.**
>
> Thank you for the invaluable suggestions.  Your suggestions have led to precious insights. We have added the following ablation experiments in Appendix A6 of the revised paper:
>
> a. Dual model (with dual constraint):  (eq6) $E_{dual}[log p(X) + log⁡p(X|y)] + log⁡p(y|X)$
>
> b. Without dual constraint: $log p(X) + log⁡p(X|y) + log⁡p(y|X)$
>
> c. Without prior: $log⁡p(X|y) + log⁡p(y|X)$
>
> d. Only backward:  $log⁡p(X|y)$
>
> The results in table 5 (Appendix A6) show that every component of the dual loss contributes to the performance positively. For top 1 accuracy: (a) dual 67.7% ;  (b) 67.0%;  (c) 66.1%; (d) 60.9%. Comparing (a) and (b) is the contribution of dual constraint, which is 0.7%. It is harder to improve upon a higher accuracy range (i.e. (b) ) than a lower range (i.e. (d) ).
>
> **Q2: Please cite relevant papers**
>
> Thank you for the great suggestions. We have cited all the suggested papers in the revised version.  Segler Nature 2018,  Segler & Waller 2017,  Strieth-Kalthoff et al 2020,  Johansson et al 2020, Yan et al 2020, Somnath et al 2020, Guo et al 2020, Coley et al Science 2019; Arus-Pous et al 2019.
>
> In our first edition, we have already cited Segler & Waller 2017 (“NeuralSym” in Table 1), and Segler Nature 2018  (Introduction).   Segler Nature 2018 focuses on multiple-step retrosynthesis. We focus on the single-step retrosynthesis, which is a critical building block of the multi-step process. We have included a discussion about Segler & Waller 2017 in the result section. We will include more discussions with other methods in the camera-ready edition.
>
> **Q3: Table 3. With Type, do you mean the reaction type is given?**
>
> Yes
>
>
> **Q4:  eq13 should be a centered dot for multiplication, not a dot.**
>
> Yes. Thanks. Fixed.
>
>
> **Q5: Please change fonts for your figures.**
>
> Apologize. We changed the fonts to “courier”.
>
>
> **Q6: Not SOTA, compared with RetroXpert (Yan et al,  NeurIPS 2020) and GraphRETRO (Somnath et al, 2020) [from other reviewers].**
>
> Please kindly refer to the response to all reviewers with the title "Reply to R1, R2, R3, R4: Not SOTA...”.
>
> We have a different setup. RetroXpert and GraphRETRO are not template free methods. The lower results are due to the fact that we are not using extra chemical features. If we add those features to the dual model, we get better results than RetroXpert.  Dual: 70.1%; 73.2% RetroXpert: 65.6%; 70.4%  (Table 4 in Appendix A. 4 of our revised paper).

---

> > ### Comment · AnonReviewer1 · 2020-11-25
> > **thank you!**
> >
> > Thank you for the additional experiments, this makes it easy to see that the dual constraint has a contribution to the increased performance.
> >
> > Regarding your answer on SOTA/not SOTA, I believe there is still a bit of confusion. I get the impression the authors have not worked with template-based methods before, because they are actually very easy to use and setup in an automatical way. I will more extensively reply in the global thread.
> > In short: It does not matter whether a method is template-free or not  (we can treat it as a black box that takes the product as input), what matters is empirical performance on the test set (and of course that overall the method produces sensible results beyond the test set, but that is harder to measure). And in that sense, this paper operates under the exact same setup as retroXpert, GLN, etc.
> >
> > I think this issue needs to be rectified, and the actual SOTA acknowledged, before the paper can be accepted at ICLR.
> >
> > I think the main purpose (and value) of this paper is the analysis using energy-based models (and I already said that I am willing to recommend acceptance solely for that), but the current discussion unfortunately distracts from that.

---

### Official Review · AnonReviewer2 · 2020-10-27
**Clarity needs to be improved**

**Rating:** 5
**Confidence:** 4

**Review:**

### Summary of the paper
This paper proposes an energy based model (EBM) for retrosynthesis. The best model (dual model) leverages the duality of retrosynthesis and reaction prediction. The EBM contains three factors: prior on reactants $p(X)$, forward reaction probability $p(y | X) and backward posterior $P(X|y)$. The duality loss penalizes the KL divergence of the two directions. The forward predictor is trained on a mixture of original data and samples drawn from the backward predictor. The dual model shows improvement over template-based and template-free baselines.

### Strength
1. The dual model is novel and interesting. EBM provides a unified view of forward and backward reaction prediction. It will be interesting to see how this improves the forward prediction performance.
2. EBM is a flexible framework, which can be applied to both template-based and template-free approaches.

### Weakness (and questions)
1. Clarity: The paper describes EBM view of many methods, ranging from sequence and graph based methods. As a result, each subsection is too sketchy and lots of details are missing. For example:
 1. For Dual-TB (ours), what is the exact model architecture? I understand that the candidates are generated from reaction templates, but what's the parametrization of $p_\alpha, p_\gamma, p_\eta$? Is it GLN?
 2. For Dual-TF (ours), what is the model architecture? Is it transformer?
 3. What is the augmented USPTO 50k? Is it randomized SMILES? For Dual-TB (ours), is it trained on augmented USPTO 50k? Why SMILES randomization matters for templated based methods?
 4. $X$ is a set of compounds, how do you generate a set of compounds in an order-invariant manner?
 5. For duality constraint, you sample from backward predictor using beam search. Why not the other way around? Why not sample from forward predictor and train your backward predictor on the additional samples?

2. At inference time, it is hard to sample from EBM. Therefore, authors propose to rank the candidates generated from other models (reaction templates or transformers). This really limits the performance (and applicability) of the approach. To my knowledge, you can do MCMC based on Langevin dynamics to sample from EBM. Is this not possible for retrosynthesis?

3. The best model is this paper actually performs much worse than the state-of-the-art models. For instance, RetroXpert (Yan et al., NeurIPS 2020) achieves 65.6% top-1 accuracy (reaction type unknown) and 70.4% top-1 accuracy (reaction type known). This is much better than the dual model (55.2%, 67.7%). Somnath et al, 2020 also achieves 64.2% top-1 accuracy (reaction type unknown), which is much higher than dual model (55.2%).

### Overall evaluation
I vote for weak reject of the paper, primarily due to the weaker result and lack of clarity. I believe the dual formulation can be applied to these above state-of-the-art models (if code is available). I suspect the weak result is primarily due to the base model (e.g., transformer). I am happy to adjust my score if there are stronger results and the clarity can be improved. One suggestion for clarity is to put perturbed / bidirectional models into appendix since they are not helpful anyway...

### Post Rebuttal
I would like to thank authors for their response. I think the paper needs to be improved further to get accepted. I do believe that the proposed dual learning method is promising, but empirical evaluation is still lacking. So my review score stays the same.

[1] Yan et al., RetroXpert: Decompose Retrosynthesis Prediction like A Chemist, NeurIPS 2020.

[2] Somnath et al., Learning Graph Models for Template-Free Retrosynthesis, 2020

---

> ### Author Response · Authors · 2020-11-24
> **Reply to R2**
>
> Thank you for the thoughtful comments. We uploaded the revised paper and changes are highlighted in blue.  Please see our response to “Weakness”:
>
> **Q1, 2, 3**:
>
> The model architecture for the dual model under template-based or template-free setup are both transformers (not GLN) (See the paragraph under eq4 on page 4). $p_{\alpha}$, $p_{\gamma}$, $p_{\eta}$ are transformer parameters.
> Dual-TB and Dual-TF have the same training procedure (including model architecture), but different inferences. In other words, we train the same dual model in TB or TF set up, which are used to rank a list of reactant candidates, e.g.  $[X1, X2, X3 .. ]$ that are associated with a given product $y$ during inference. The proposal reactant list can be generated with templates or without templates, leading to template-based or template-free methods.
>
> **Q4**:
>
> Yes.  The augment procedures of USPTO50K are as follows. For each reaction,
> 1. Replace each molecule in reactants or product using random SMILES.
> 2. Random permute the order of reactant molecules.
>
> We updated it in the revised draft.
>
> We trained Dual-TB on both USPTO 50K (Table 2 upper) and augmented USPTO 50k (Table 2 bottom).
>
> Random SMILES matters for “sequence-based model”, as opposed to “graph-based model”.  Sequence-based models linearize the 2d molecule to a 1d sequence. The linear compression losses a rich local environment of a molecule. Random SMILES are different linearizations of the same molecule, which rescue some of the information loss. Using random SMILES also prevents transformers from overfitting. Note that as we don’t add new molecules, random SMILES augmentation won’t influence graph-based models’ performance, as graph-based models are permutation invariant.
>
> Random SMILES doesn’t have a distinguished effect on template-based versus template-free models -- it benefits both template-based and template-free models -- as long as the model is sequenced based (Table 2 template-based vs Table 3 template-free).
>
> **Q5**:
>
> There are two ways:
> 1. Data level (we used):
> As stated above, the Random SMILES data augmentation we used includes permuting the order of reactants, which achieves order invariant during training stochastically.
>
> 2.  NN level: Modify the position encoding in transformer implementation:
> Transformer has a position encoding to mark the different locations on an input sequence. We modified the position encoding such that each molecule starts with 0 encodings, as opposed to the concatenated position in the reactants sequence.  We didn’t use it in our paper, because (1) data level  has superior performance and is easy. See details in Appendix A12. Table 7 of revised paper.
>
> **Q6**:
>
> Yes, we can draw samples from the forward model and train backward on the additional samples. We can also do both (the paper plus this suggestion), or even repeat them a few times.  However drawing samples from the forward direction would either require empirical reactants from data, or from the learned $p(X)$, which is harder than drawing samples from the backward direction.
>
> **Q7**:
>
> It is true drawing samples from EBMs can be generally challenging. Langevin requires differentiability w.r.t the observations, i.e., molecules, which is NOT applicable here since the molecule SMILES string is discrete. Fortunately, using proposal distribution to sample from EBM is a principled way in general, such as the Metropolis Hastings algorithm. In our case, using the $p(X|y)$ as a proposal is a good trade-off, as it has full support in the molecule space, while being close to the EBM distribution. In practice, $p(X|y)$ can be efficiently obtained by K-size beam search.
>
> **Q8: Not SOTA compared with RetroXpert and GraphRETRO**
>
> Please see the response to all reviewers with the title "Reply to R1, R2, R3, R4: Not SOTA...”.
>
> **Q9: The dual formulation can be applied to these above state-of-the-art models (if code is available).**
>
> Yes. The generalization is an advantage of the dual framework, as it can be adapted to different models. RetroXpert's code is not uploaded yet: https://github.com/uta-smile/RetroXpert (only data is uploaded. Not code). We are happy to adapt RetroXpert into the dual model framework if the authors upload code.
>
> **Q10: Are weak results due to transformer?**
>
> See the response titled"reply to all; NOT SOTA ..." and Table 4 in Appendix A. 4 of our revised paper.  The weak results are due to that we are not using extra chemical features. If we add those features to the dual model, we get better results than RetroXpert.  Dual: 70.1%; 73.2% RetroXpert: 65.6%; 70.4%
>
> **Q11: Move perturbed / bidirectional to appendix**
>
> Good point. Thanks. But our goal is to present a full picture of different model designs. In our opinions, both good and bad performances are valuable to trigger helpful discussions and inspire better models in the community.  But since we run out of space, we plan to move some parts into the appendix.

---

### Official Review · AnonReviewer4 · 2020-10-30
**Review of Energy-Based view of Retrosynthesis**

**Rating:** 8
**Confidence:** 4

**Review:**

SUMMARY

This paper uses the statistical physics-inspired energy-based model formalism to study the by now "canonical" problem of retrosynthesis using deep learning. The authors use an interesting variant that combines forward and backward prediction. The authors use template-based and template-free models.


PROS

- This reviewer believes that energy-based models have an elegance and connection to statistical mechanics that should be explored more in the area of machine learning. This work goes in this interesting direction.
- Based on the above, and as far as the reviewer is appraised, this is a unique, non-derivative direction in the field and therefore deserving of consideration for acceptance.
- The dual model seems to be very useful given the increase in template-based and non-template-based model performance. This could be applied to other transformer-based tasks in chemistry and graph-based ML
- The authors compared their models to a variety of SOTA models and approaches, they also were thorough and explored both DeepSMILES and SELFIES.


CONS
- Some of the mathematical formalism could be moved to supplementary to allow for better discussion.


MINOR FORMATTING
- The authors may want to give the manuscript a pass for grammar. There are missing articles in a few sentences.

---

> ### Author Response · Authors · 2020-11-24
> **Reply to R4:**
>
> Thank you for the helpful suggestions. We thank the reviewer for the positive feedbacks and for highlighting the advantages of our work. We have uploaded the revised paper, and the changes are highlighted in blue.
>
> “Moving some mathematical details into supplementary” is a very helpful suggestion,  as it can (1) make the paper more readable, (2) allow in-depth discussion, and (3) avoid distracting the readers from unnecessary details.
>
> We plan to do so for the camera-ready version. However, currently, we are debating on which parts should be put into the appendix, as different readers seem to be interested in different parts.

---

### Author Response · Authors · 2020-11-24
**Reply to R1, R2, R3, R4: “Not SOTA. RetroXpert  and GraphRETRO have better performance”.**

We upload the revised paper and changes in blue.

# Short answer:

We have a **different experimental setup** from RetroXpert or GraphRETRO. RetroXpert and GraphRETRO use additional important chemical information (e.g. reaction centers, and other handcrafted template-related features) during modeling and training, whereas we do not. Adding useful chemical features usually has a ~10% improvement in accuracy (for example, “reaction type” is a well-known example that improves ~10% acc regardless of model designs). Because of the usage of chemical features, we classify RetroXpert and GraphRETRO as **semi-template based methods**, whereas our models as **template-free models** (See Appendix A1, 2, 3, 4 of revised paper). **So comparing our models with RetroXpert or GraphRETRO is not a head-to-head comparison**.

To prove the above point further, we performed experiments to include these features in our model. The results showed these features boost our model significantly, as expected. Assuming the reaction center is known (due to time constraint of the rebuttal), the dual model improves to **70.1% from 54.5%** without reaction type, and improves to **73.2% from 66.2%** with reaction type. See appendix A.4 Table 4 of the revised version.  (70.1%, 73.2%) is higher than RetroXpert (65.6%, 70.1%). However, our paper focuses on the comparison of different models under the same setup (template-free model without any other additional chemistry features).  Adding extra features to improve accuracy is not our goal here. Apologies for not making this point clear.

# Detailed answer:

This paper aims at understanding different modeling designs under the setup of full-automatic template-free, which are defined as models without using hand-crafted chemistry features, including but not limited to templates, sub-parts of templates, reaction centers, and atom mapping (matching of the same atoms in reactants and product).  We are interested in this setup because (1) those handcrafted features are often not available for real-world large datasets. For example, without atom-mapping, reaction centers have to be manually labeled or inferred from substructure searches and/or predictive models, which may be imperfect. With atom-mapping, reaction centers can be extracted computationally. However, getting good atom mapping for large datasets is an equally challenging problem. (2) The future of retrosynthesis is to train models that can think beyond existing rules (e.g. template), and learn useful things without using prior human knowledge.

**RetroXpert and GraphRETRO use lots of useful chemistry features to boost accuracy**

1. RetroXpert  and GraphRETRO both use “reaction centers”, which are almost equivalent to templates.

Although RetroXpert and GraphRETRO claim they are template-free methods (for not explicitly performing template-matching), they are not fully automatic template-free according to our standard above. Similar with G2G (Shi et al ICML 2020), RetroXpert and GraphRETRO use “reaction centers” as additional information to supervise their algorithm during training ( $y_{ij}$ in eq2 for RetroXpert paper;  $y_{uvk}$ in eq5 for GraphRETRO paper). The “reaction centers” preserve almost equivalent information as templates-- once reaction centers are given, the product can be broken into two parts (denoted as “synthons”). The templates can be recovered by removing nonessential atoms from products and synthons (See GLN Figure 1, Dai et al NeurIPS 2019). Therefore, we denote RetroXpert or GraphRETRO as semi-template based methods.

2. Besides reaction centers, what other template-related features do RetroXpert or GraphRETRO use?

- **RetroXpert**: adds “the product side of templates'' as atom features (See sec “Atom and bond features” on page 5 of RetroXpert paper). These features take 654 dimensions out of a total of 791 atom features (Appendix B).  Ablation study shows without  “product side of template", the accuracy for identifying centers drops by 2% with reaction type, and 5% without reaction type (D.2 in Appendix) . The accuracy of the retrosynthesis may decrease further. In conclusion, these “product side of templates” significantly improves accuracy.
- **GraphRETRO**:  predefines a list of leaving groups (Sec 3.2 of GraphRETRO paper). GraphRETRO constrains the leaving groups to be a list of predefined groups extracted from the entire database (including test data. page 5, line 5, “ a standard dataset with 50, 000 examples”), which allows training to use test data. Additionally, doing so simplifies the task from a generative task (of reactants) to a classification problem, which is not able to generalize to unseen molecules.


Reference
[1]  Yan. et al. (RetroXpert):
NeurIPS 2020. https://arxiv.org/abs/2011.02893
[2]  Somnath et al. (GraphRETRO)
https://arxiv.org/abs/2006.07038
[3] Shi et al. (G2G): ICML 2020. https://arxiv.org/pdf/2003.12725.pdf

---

> ### Comment · AnonReviewer3 · 2020-11-24
> **My confusing in "hand-crafted**
>
> Thank you for your feedback.
>
> I'm getting confused with your answer, so would you please help me understanding the statement more precisely.
>
> "These handcrafted features are often not accessible for real world application" (Appendix A.3)
>
> My understanding is that no atom mappings in large datasets are fully manually maintained.
> The USPTO-50K dataset is also equipped with the machine-computed atom mapping.
> I mean, any new dataset in the wild we can apply a mapping script and get atom-mapping and reaction centers, whose qualities are roughly same with the atom map of USPTO-50K. Then, I understand the atom-mapping and the reaction centers are in fact accessible for real-world studies. Is this understanding wrong?

---

> > ### Author Response · Authors · 2020-11-24
> > **Reply to R3: My confusing in "hand-crafted"**
> >
> > Thank you for your prompt reply and helpful question!
> >
> > ### Q1: Can researchers run an atom mapping algorithm on any real-world dataset and get good atom mapping with the same quality as USPTO50K?
> >
> > Not necessarily. Atom mapping algorithms do better on simple reactions, and worse on complex reactions. Real-world datasets are more likely to contain complex reactions. The accuracy of atom mapping on real-world datasets drops. In our opinion, atom mapping is in general a quite hard problem and an atom-mapping algorithm may not be perfect.
> >
> > https://chemrxiv.org/articles/preprint/Atom-to-Atom_Mapping_A_Benchmarking_Study_of_Popular_Mapping_Algorithms_and_Consensus_Strategies/13012679/1
> >
> > This paper is a benchmark study of popular atom mapping algorithms on a dataset with human-labeled atom mapping ground truth. See table 1, the best algorithm RXNMapper (transformer-based atom mapping algorithm) achieves 83%, which is not perfect. Even with the best algorithm RXNMapper, the authors observed undesired performance (page 15):
> >
> > >“when applied to the entire USPTO dataset errors appear. For example, out of 541 reactions of reduction of carboxyl groups to alcohols found in USPTO only 58 were mapped correctly, while 483 had wrong AAM. For reactions of methyl ester formation using methanol 493 reactions were correct but 225 were found to be incorrectly mapped by RXNMapper.”
> >
> > **Additionally, since “template” or “reaction center” can be extracted from “atom mapping”, we lean to characterize models using “atom mapping” as “semi-template-based” methods (or more precisely, 'mapping-based' methods). Semi-template-based methods are not the focus of our paper.**
> >
> >
> > ### Q2 Reiterate key points: We don’t want to let the accuracy gain due to incorporating additional chemical features mislead (or interfere with) the evaluation of different modeling.
> >
> > We want to reiterate the key point of this response titled “Reply to R1,2,3,4 NOT SOTA ...” is that we want to make **a fair comparison** where every model gets access to the same information. We don’t want to let the accuracy gain due to incorporating additional chemical features **mislead the evaluation of different modeling**. In this paper, we are mainly interested in methods without using chemistry features, especially template-related features (reaction center, product side of template, templates, etc). We hope eventually the models can learn all this chemistry knowledge by themselves, instead of building on the output of another algorithm, say atom mapping algorithm.
> >
> > Let us know if it helps. We are happy to answer more questions. Thank you for the time.

---

> ### Comment · AnonReviewer2 · 2020-11-24
> **Experimental setup is confusing**
>
> Thank you for your feedback. Unfortunately I am also confused with your answer.
>
> 1. You claim that "RetroXpert and GraphRETRO use lots of useful chemistry features to boost accuracy". Doesn't Dual-TB model also uses reaction templates? Compared to atom mapping, I think reaction templates are richer chemistry features. Besides, reaction templates are extracted with ground-truth atom mappings provided in USPTO. So inevitably you are using atom mapping and reaction centers in the training set. From the updated paper, dual-TB performance is still lower than RetroXpert and GraphRETRO.
>
> 2. Your template-free model achieves 70.1% top-1 accuracy when using reaction centers. Unfortunately this is not a head-to-head comparison with RetroXpert and GraphRETRO because they need to predict the reaction center and then generate precursors. Moreover, reaction centers are computed using true atom mapping. So dual-TF is no longer template-free in this setting. In fact, you are using stronger features.

---

> > ### Author Response · Authors · 2020-11-24
> > **Re R2: "Experimental setup is confusing"**
> >
> > We are sorry for the confusion. We appreciate both of your great questions. Please see our clarification below. We uploaded a new revision with one edit marked in magenta on page 13.
> >
> > **Q1: “Dual-TB (Table 1) performance is still lower than RetroXpert and GraphRETRO.”**
> >
> > We did not have time to touch template-based experiments (Dual-TB). We did not do experiments on adding reaction centers to Dual-TB (Dual-TB part is not updated). We only use Template-free setup to show the effect of incorporating “reaction center”.  But based on our previous experiences under the same setup, Dual-TB usually performs similar with Dual-TF but a bit better, as they have the same training procedure (Only difference between the two are how to propose candidate list during testing). Our guess is Dual-DB with reaction center known should be similar with Table 4 Appendix A4, if not a bit better.
> >
> > To be clear, Dual-TB does not use template or reaction centers during training. Dual-TB only uses “templates” to propose a list of candidates during testing. So Dual-TB is template-free during training.
> >
> > We appreciate RetroXpert and GraphRETRO’s impressive results. Because of their great results and our A4 appendix results, we are convinced that compared with directly matching templates (“template-based methods”), reaction centers actually are more accessible/easier features for the algorithm to learn (directly points the key information without introducing noise from templates). Just for this paper, we are more interested in the modeling part of retrosynthesis than chasing SOTA by using additional information during training.
> >
> > **Q2:  “Your new experiments in Appendix A4 (70.1% and 73.4%) are NOT head-to-head comparisons with RetroXpert or GraphRETRO”.**
> >
> > You are absolutely correct -- Appendix A4 experiments are using “true reaction centers”, but RetroXpert or GraphRETRO has to infer them, hence not head-to-head comparison. We apologize for the misunderstanding; **our intent was simply to demonstrate that adding this information would improve performance (given as an upper bound) rather than to serve as a head-to-head comparison. We think the upper bound is still informative to show the features are critical for improving accuracy.** As we stated in Appendix in our previous edition (third to last paragraph).
> >
> > > “Although in the above experiments, we are given the reaction center, that is we do not infer them, Table 4 still serves well as a proof of concept that incorporating useful handcrafted chemistry features can significantly boost accuracy regardless of model design.”
> >
> > Due to the limited time of rebuttal, we do not have time to infer the reaction center first and use the inferred ones to our experiments. **We can consider 70.1% and 73.4% to be upper bounds on our experiments when we can infer the reaction center perfectly**; we made this clear in the newly revised paper. However, we would like to also emphasize that inferring centers (using additional information “reactor center” as supervision during training) is not the direction we would like to pursue as these models rely on more information than the “full automatic template-free” models we are proposing here. The experiments we did in A4 appendix are just to alert the community and also prove to reviewers that we should compare models with the access of same information, not to claim SOTA.
> >
> > **Q3: “With true reaction center, your Dual-TF with true reaction center given (70.1%, 73.2%) is not template free anymore”**
> >
> > Sorry for not making the following points clear in the rebuttal, in our revised paper:
> > **We do not claim 70.1% is the performance of our Dual-TF**. Please kindly refer to our previous revised paper (Table 1), 55.4% is still our evaluation (in black). We put 70.1% in the appendix, not the main text, as an “auxiliary study” proof of concept that gives an upper bound on performance for the case where we are incorporating reaction centers as additional features.   We think the upper bound is still informative to show the additional features are critical for improving accuracy.
> >
> > We did not have time to do experiments on adding other chemistry features to our dual model, such as “product side of template” which are demonstrated to be useful in RetroXpert.  Doing so may boost accuracy for dual-semi-template-based methods even further.
> >
> > Many of the above experiments we did not do is due to rebuttal's time constraints, and also more importantly they are not the main focus of our paper: full automatic template free methods.
> >
> > Thank you again for the great review and questions! We are happy to answer more questions.

---

### Decision · Program_Chairs · 2021-01-07
**Final Decision**

**Decision:**

Reject

**Comment:**

Before the discussion phase nearly all reviewers had doubts about the comparison of the current work with state-of-the-art works (notably Yan et al., 2020, RetroXpert, and GraphRETRO). The authors then compared with these works and emphasized that these works rely on hand-crafted features. They argue that the fairest comparison is the one where each method uses the same sort of features during train/test time. This is because in certain real world settings we may not have accurate estimates of such features (e.g., atom mappings, templates, reaction centers). However, in the revised version of the paper the authors did not adhere to this concept of fair comparison in Table 4 of Appendix A.4. Here their method uses reaction centers as input while baselines do not. While the authors claimed that the comparison here was designed to show how reaction centers provided as input improved performance, this doesn't seem like a good way to show it: to isolate the improvement due to reaction center inputs you should fix everything else, i.e., the rest of the method.

Apart from the above contradiction, I buy the arguments of reviewers that distinguishing between methods that use hand-crafted features and those that do not is not a meaningful distinction. One can apply atom-mapping or reaction center discovery algorithms as data preprocessing before applying other methods. Ablation studies where such preprocessing is added or removed are interesting, but it is completely fair for any method to use such preprocessing before applying their method, it is up to the modeller.

I would have argued for acceptance had the authors either (a) just included results from SOTA methods (one, RetroXpert was published 1 month after the ICLR submission deadline), and/or (b) reran their approach with such preprocessing. However the authors ended up hurting the submission by emphasizing a difference between using handcrafted features and not, then contradicting their experimental setup in Table 4.

This is a good paper, but I agree it is not ready to be accepted at ICLR. I recommend the authors do the following: (a) use any preprocessing they want for their method and compare with the state-of-the-art, (b) if they want they can run their method without any preprocessing as an interesting ablation study, (c) remove Table 4 (as (b) already does this type of an ablation study), (d) describe recent work through the lense of EBM, (e) resubmit to a strong ML conference. The new submission will be much stronger.